# The Fabrication and Properties of a Bendable High-Temperature Resistance Conductive Pitch-Based Carbon/CNT Film Nanocomposite

**DOI:** 10.3390/nano11030758

**Published:** 2021-03-17

**Authors:** Zhe Che, Shaokai Wang, Yizhuo Gu, Wei Zhang, Cai Jiang, Min Li

**Affiliations:** 1Key Laboratory of Aerospace Advanced Materials and Performance (Ministry of Education), School of Materials Science and Engineering, Beihang University, No. 37 Xueyuan Road, Haidian District, Beijing 100191, China; chezhe1995@sina.com (Z.C.); benniegu@buaa.edu.cn (Y.G.); permitivity90@163.com (W.Z.); jane19871025@163.com (C.J.); 2Ningbo Institute of Technology, Beihang University, Ningbo 315800, China

**Keywords:** carbon nanotube film, mesophase pitch, carbonization, multi-functionality

## Abstract

This paper fabricates a carbon nanotube (CNT ) film-reinforced mesophase pitch-based carbon (CNTF/MPC) nanocomposite by using a hot-pressing carbonization method. During the carbonization, the stacked aromatic layers tended to rearrange into amorphous carbon, and subsequently generated crystalline carbon in the matrix. The continuous entangled CNT networks were efficiently densified by the carbon matrix though optimized external pressure to obtain the high-performance nanocomposites. The CNTF/MPC@1300 displayed a stable electrical conductivity up to 841 S/cm at RT-150 °C. Its thermal conductivity in the thickness direction was 1.89 W/m∙K, an order of magnitude higher than that of CNT film. Moreover, CNTF/MPC@1300 showed a mass retention of 99.3% at 1000 °C. Its tensile strength was 2.6 times the CNT film and the tensile modulus was two orders of magnitude higher. Though the CNTF/MPC nanocomposites exhibited brittle tensile failure mode, they resisted cyclic bending without damage. The results demonstrate that the CNTF/MPC nanocomposite has potential application in multi-functional temperature resistance aerospace structures.

## 1. Introduction

Carbon reinforcement/carbon matrix (C/C) composites possessing a whole carbon system with excellent stability, extraordinary electrical and thermal conductivities, outstanding ablation resistance, and superior friction performance [1,2] are widely applied in the fields of space aircrafts, automobiles, and biomedical devices, such as nose tips, rocket nozzles, disk brakes, and artificial bone [3,4,5]. To advance technologies and applications in related fields, carbon nanotubes (CNTs) as high-performance nanoscale reinforcement have also been used to attempt to reinforce carbon matrix. Single CNT has a relatively low density at around 1.3 g/cm^3^ with very high tensile strength and modulus of 1.0 TPa and 50 GPa, respectively [6,7,8]. Meanwhile, the extraordinary electrical and thermal conductivities of CNTs may also impart great physical properties to CNT/C composites [9,10,11]. In addition, CNTs may be manipulated via various methods to achieve diverse microstructures in the final nanocomposites [12,13,14,15], and the properties of CNT/C composites can be easily adjusted to better meet the needs of different applications. Therefore, CNTs could prove to be the answer for the most promising reinforcements of the next generation of C/C composites.

CNTs were first applied in CNT/C bulk composite as fillers by means of mechanical mixing [16,17]. However, CNT powders are prone to aggregating. This limits the addition of content to CNTs within the CNT/C composites due to poor dispersion, stability of the CNT suspension, and processability during the combination process with the resin. On the contrary, as a kind of 2D macroscopic structure, CNT film could avoid random aggregation and difficult dispersion of nanotubes, providing non-damaged native structure and a high load of nanotubes for fabricating CNT/C composites. Therefore, extending the potential high mechanical properties of CNT into macroscopic materials has become a research focus in recent years.

As regards the fabrication of C/C composites, chemical vapor infiltration (CVI) is frequently used [18,19]. Jin et al. [20] prepared CNT/C composites by using the CVI method, and the compressive strength and electrical conductivity reached 148.6 MPa and 191 S/cm, respectively. Applying a similar method, Gong et al. [21] fabricated CNT/C composites with CNT arrays, and the electrical conductivity of the final CNT/C composite reached 161 S/cm in the aligned direction. Faraji et al. [22] used multiple cycles of CVI and a graphitization process to obtain a CNT/C composite with a larger graphite region and higher thermal oxidation stability. Li et al. [23] further accelerated the manufacturing of CNT arrays/C composites by a simple step with the combination of the chemical vapor deposition and CVI processes. Another popular method, the precursor infiltration pyrolysis (PIP) method, has also been widely used to prepare high-loading CNT nanocomposites, in which CNTs are densified by repeated precursor impregnation and carbonization. Park et al. [24] produced CNT/pitch-C composites through three cycles of the PIP method and the electrical conductivity increased from 200 S/cm to 425 S/cm. Zhang et al. [25] reinforced CNT film with carbon matrix by a double infiltration pyrolysis process to enhance the tensile strength sixfold. Stein et al. [26] attempted CNT forests to fabricate CNT/C composites via a vacuum-assisted wetting PIP process, in which the matrix porosity remained at 55% after the second infusion and pyrolysis. To further improve the infiltration of carbon matrix, Zhu et al. [27,28] fabricated the aligned CNT/C composite by spraying the precursor solution of carbon matrix onto a continuous CNT sheet and winding it onto the rotating mandrel.

However, the quality control of CNT/C still remains a challenge due to the tightly packed CNTs and limited space for matrix impregnation. The carbon matrix tends to form a shell on the surface of the composites, and cracks and pores inevitably form in the CNT/Cs during the fabrication process [29]. Additionally, the time cost and energy consumption are relatively high for the abovementioned CVI and PIP methods. The continuous CNT films are appropriate for fabricating high-performance CNT/C composites with a super high content of CNTs, which plays a key role as the reinforcing and functional constituent, but the evolution of the fine structures and properties of the composites requires deep understanding.

Herein, a flow catalyst chemical vapor deposition (FCCVD) CNT film combined with a pitch-based carbon nanocomposite was investigated. The CNTs inside aerogel are relatively straight, with a length close to 1 mm, and randomly orient, entangle, and connect into a continuous network. These structure features ensure excellent mechanical and electrical properties as well as good processability. To reduce the time and energy cost, a hot-pressing carbonization process was developed. The hot-processing parameters were optimized to avoid the probable generation of a resin shell to obtain a complete infiltration of mesophase pitch into the entangled CNT network and to suppress possible void formation without using cyclic steps. The hot-pressed CNT film/pitch was then carbonized under high temperatures to prepare the CNTF/MPC nanocomposite. The mechanical properties, heat resistance, and electrical and thermal conductivities of CNTF/MPC were measured to evaluate the nanocomposite comprehensively. Moreover, FTIR, Raman, and X-ray diffraction tests were probed to reveal the structure evolution of carbon matrix during the carbonization process.

## 2. Experiment

### 2.1. Materials

Carbon nanotube film (CNTF) synthesized by the FCCVD method was purchased from Suzhou Institute of Nano-Tech and Nano-Bionics, Chinese Academy of Sciences. The as-received CNT film was made up of entangled multiwall CNTs [30] with a thickness of approximately 20 μm and a bulk density of 0.37 g/cm^3^. Mesophase pitch, which had a density of 1.2 g/cm^3^ and a sulfur content of 2–5 wt.%, was provided by Guangzhou Tongda Chemical Engineering Co., Ltd. The softening point of the mesophase pitch was between 255 and 295 °C.

### 2.2. Fabrication of CNTF/MPC Nanocomposites

Figure 1 illustrates the fabrication process of the CNTF/MPC composite. The as-received mesophase pitch was first dissolved in xylene, and the pitch solution was applied to the surface of the CNT film with a mass ratio of pitch to film equal to 2:1. The solvent was then vaporized at ambient temperature for 12 h. Subsequently, the pre-impregnated CNTF/MP (mesophase pitch) composite was hot pressed at 360 °C and 1 MPa for 2 h, with a stabilization process at 220 °C under an air atmosphere [31]. Furthermore, the stabilized CNTF/MP composite was placed in a tubular furnace, and the carbonization process was carried out under an argon atmosphere with a gas flow rate of 300 sccm/min. The heating ramp and cooling rate were controlled at 2 °C /min and 4 °C /min, respectively. The CNTF/MP composites were treated separately at 1000 °C and 1300 °C for 30 min [32], and the received CNTF/MPC samples were denoted as CNTF/MPC@1000 and CNTF/MPC@1300, respectively.

### 2.3. Material Characterizations and Testing

The tensile properties were tested on an Instron 3344 tensile testing machine (Instron Inc., Boston, MA, USA) according to ASTM 3379. The specimen size for tensile test was 50 mm × 1 mm, and five specimens were measured for each sample. Thermogravimetric analysis (TGA) was performed by using STA449F5 (Netzsch Inc., Bavaria, Germany) with a ramp rate of 10 °C/min during air or nitrogen flow. Differential scanning calorimetry was tested by Mettler TGA/DSC 3^+^ (METTLER TOLEDO Inc., Zurich, Switzerland) with a heating rate of 5 °C/min during nitrogen flow. Electrical resistance was measured using the two-probe method with a TH2512 DC low-resistance tester (Tonghui Inc., Jiangsu, China). Silver paste was coated on the two side edges of the sample as electrodes. Thermal diffusivity was measured by LFA467 (Netzsch Inc., Bavaria, Germany) at 25 °C. The microstructures of the CNTF/MPC nanocomposites were characterized by using a JSM7500 (JEOL Inc., Tokyo, Japan) field emission scanning electron microscope (SEM) operated at 3.0 kV and a JEM2100 (JEOL Inc., Tokyo, Japan) transmission electron microscope (TEM) with an accelerating voltage of 200 kV. The cross-sections of the CNTF/MPC nanocomposites were characterized by a focused ion beam scanning electron microscope (FIB-SEM) operated at 5.0 kV (Helios G4 PFIB HXe, Thermo Fisher Scientific, Waltham, MA, USA). Raman spectra were measured by using a micro confocal laser Raman spectrometer (LabRAM HR Evolution, Horiba scientific Inc. Palaiseau, France) with a He–Ne laser excitation line (633.0 nm) and a spot size of 1 μm^2^. Infrared spectra were analyzed using a Nicolet-6700 Fourier transform infrared spectrometer (Thermo Fisher Scientific, Waltham, MA, USA). X-ray diffraction (XRD) measurements were carried out using a XRD system (D8 Advance, Bruker Nano Inc., Munich, Germany) with Cu Kα radiation (λ = 0.15406 nm) at a scanning velocity of 5°/min.

## 3. Results and Discussion

### 3.1. Mechanical Properties of CNTF/MPC Composite

Figure 2 shows the typical stress–strain curves of CNT film and the corresponding CNTF/MPC nanocomposites. As expected, the stress–strain curve of CNT film showed a linear relationship at first, and then tensile inflection occurred during the second stage with a strain of 1–3%. Subsequently, the tensile stress increased slowly until the breakage of the CNT film at a strain of 19.8%. The resultant tensile strength and modulus of CNT film were only 45.5 MPa and 0.19 GPa, respectively, and the fracture energy reached 15.2 J/g. This high energy absorption characteristic was attributed to the easy slippage and alignment nanotubes in the entangled CNT networks due to the weak inter-tube van der Waals force.

According to the traditional fabrication technology of pitch-based carbon fiber, the mesophase pitch containing stacking aromatic layers [33] rearranges into carbon matrix mainly composed of amorphous carbon in 1000 °C heat treatment. Consequently, 1000 °C heat treatment was selected to carbonize the CNTF/pitch composite. In Figure 2 we can see that the tensile stress–strain curve of CNTF/MPC@1000 showed an obvious linear relationship before breakage at 0.62%. The tensile strength and modulus were 108.6 MPa and 18.9 GPa, respectively, significantly higher than those of CNT film. Meanwhile, the fracture energy of CNTF/MPC@1000 was only 0.22 J/g, lower by two orders of magnitude than that of CNT film. From Figure 1 we can find that the loose combined CNTF/pitch composite displayed a metallic luster after the carbonization. The measurement showed that CNTF/MPC@1000 had a volume density of 1.38 g/cm^3^, which was four times of that of pristine CNT film (0.37 g/cm^3^). With the carbonization temperature increasing to 1300 °C, the density of the CNTF/MPC nanocomposite increased to 1.53 g/cm^3^ with a higher tensile strength and modulus of 119.8 MPa and 23.8 GPa, increasing by 160% and 1240%, respectively. Moreover, the tensile fracture energy was only slightly decreased to 0.2 J/g, which was equivalent to that of CNTF/MPC@1000. Further cyclic bending tests showed no observable plastic deformation or damage to the glistening CNTF/MPC@1300 nanocomposite due to the great flexibility of the entangled continuous CNT network. This evolution tendency of mechanical properties was similar with the reported carbon fibers/carbon composites [34] and other types of CNT/C nanocomposites [35]. The tensile strength, modulus, and fracture energy of CNTF/MPC were also higher than the reported graphene oxide paper (42.8–75.9 MPa, 5.9–17.2 GPa and 0.19 J/g, respectively) [36,37,38].

Representative fracture morphologies of the CNT film and the corresponding CNTF/MPC nanocomposites are shown in Figure 3. The CNT film displayed a very long pull-out length of nanotubes and the sublayer delamination is ubiquitous in Figure 3a. This characteristic explains well the high fracture energy and the large breaking elongation of the CNT film, while also contributing to a relatively low tensile strength and modulus. In comparison, no visible sublayer delamination can be found in the fracture morphologies of CNTF/MPC@1000 or CNTF/MPC@1300 in Figure 3b,c, which is indicative of full infiltration of the pitch-based carbon matrix inside the nano and micron space of the CNT networks. From Figure 3d one can easily observe long, wavy CNT pull-outs from the fracture of the CNT film, whereas the lengths of CNT pull-outs are much shorter for CNTF/MPC@1000 in Figure 3e. This can be attributed to the lock-up effect of carbon matrix, which may lock the entangled CNT network, and thus restrain CNT slippage and alignment. With the increase in carbonization temperature, the CNTF/MPC@1300 demonstrated more evident brittle failure features with a smoother fracture surface and shorter CNT pull-out lengths (average 6 um) than that of CNTF/MPC@1000 (10 μm) (Figure 3f). A close observation may find a certain amount of micro-cracks in the fracture cross-sections of the CNTF/MPC nanocomposites in Figure 3b,c. This demonstrates that the carbon matrix is more brittle with a higher carbonization temperature, tensely coating around the nanotubes and leading to higher stress transfer efficiency among the networks.

### 3.2. Structure Evolution of CNTF/MPC Composite

FTIR was explored to reveal the composition change in the mesophase pitch during the carbonization process. Figure 4 shows the IR spectra of the as-received mesophase pitch (MP), and the carbonization-treated mesophase pitch at 1000 °C (MPC@1000) and 1300 °C (MPC@1300). For the IR spectrum of MP, as expected, the peaks at 3040 cm^−1^ and 1600 cm^−1^ corresponded to the aromatic C–H and C–C stretching [39], respectively. The peaks in the region of 750–870 cm^−1^ were attributed to the aromatic C–H deformation. The aliphatic C–H asymmetric and symmetric stretching was proven by the peaks at 2918 cm^−1^ and 2852 cm^−1^, respectively. In addition, the peak at 1440 cm^−1^ was due to the methylene C–H in-plane bending, which was derived from naphthenic rings in the pristine mesophase pitch [40]. After stabilization and subsequent carbonization at 1000 °C, the spectrum showed C–O groups peaking at 1050 cm^−1^. Meanwhile, the characteristic peaks of aliphatic at 2918 cm^−1^, 2852 cm^−1^, and 1440 cm^−1^ decreased evidently, which may have been caused by a thermally induced breakage of naphthenic rings, dehydrogenation, and scission of aliphatic hydrocarbon in MP [41]. The characteristic peaks of the aromatic at 3040 cm^−1^, 1600 cm^−1^, and 750–870 cm^−1^ also decreased significantly, indicating the pyrolysis and rearrangement of stacking aromatic layers. For MPC@1300 in Figure 4, the aliphatic peaks (at 2918 cm^−1^, 2852 cm^−1^, and 1440 cm^−1^) are barely observable, and other characteristic peaks cannot be detected. This demonstrates that pyrolysis of aromatic layers and aliphatic hydrocarbon took place completely during the carbonization of MP at 1300 °C. One may infer that the mesophase pitch was fully transformed into carbon matrix of MPC@1300. The weak infrared activity of MPC@1300 had a similar IR spectrum as raw graphite [42,43], demonstrating the generation of crystalline carbon with graphite-like structures in the carbonized matrix.

Raman spectra of MP and carbonized MPCs were measured to characterize the structure evolution of the mesophase pitch matrix, as shown in Figure 5a. The Raman spectra of carbon-based materials are always characterized by G band and D band at around 1580 and 1350 cm^−1^, respectively. The G band corresponds to the vibration of an SP^2^ hybridized carbon atom in a graphite layer of a hexagonal ring, whereas the D band is a vibration mode induced by an SP^3^ hybridized carbon atom or defects in graphite carbon. Hence, the intensity ratio of G band to D band was considered the characteristic parameter proportional to the crystalline degree of the carbon materials [35]. For MP the I_G_/I_D_ was 1.32, whereas the value of MPC@1000 decreased to 0.92. Mochida et al. [33] disclosed that numerous micro-domain units exist in mesophase pitch, which are formed by the stacking of aromatic layers. These types of micro-domain units of aromatic layer stacks are believed to contribute to the larger value of I_G_/I_D_ of MP. During the carbonization treatment at 1000 °C, the pyrolysis of the stacking aromatic layers reduces the size and the number of the crystalline structures, leading to a decreased I_G_/I_D_ of MP@1000. This was well confirmed by the sharp decreases in the characteristic peaks of aromatic functional groups at 3040 cm^−1^, 1600 cm^−1^, and 750–870 cm^−1^ in the IR spectra. With the carbonization temperature increasing to 1300 °C, the ratio I_G_/I_D_ of MPC@1300 then increased to 1.04, ascribing to nascent crystallite carbon within the matrix. In Figure 5a, a tiny G’ band can be observed at around 2670 cm^−1^ of MPC@1000, and the peak is very evident for MPC@1300. This could be ascribed to the secondary structure of graphene carbon formed in the carbonized MPCs, wherein the graphene was newly generated in MPC@1000 and grew much larger in MPC@1300.

Figure 5b shows the XRD patterns of MP, MPC@1000, and MPC@1300. The peak around 25° was attributed to the lattice planes of (002), and the intensity and width of (002) peak corresponded to the proportion and the size of the crystalline carbon, respectively. The peak of (10*) around 42° was the superposition of the (100) and (110) peaks due to the in-plane reflection of crystalline carbon [44]. The evaluated structural parameters are listed in Table 1. In MP, the stacking aromatic layers showed a relatively large d(002) of 0.352 nm and an L_C_ of 5.05 nm, whereas the L_C_ of MPC@1000 decreased to 2.13 nm, corresponding to the pyrolysis of the aromatic layers. Meanwhile, the d(002) of MPC@1000 decreased to 0.349 nm and the peak intensity of (10*) increased considerably, both caused by the formation of crystalline carbon during the carbonization process. For MPC@1300, the d(002) further decreased to 0.348 nm and L_C_ increased to 2.88 nm, manifesting that the graphite crystallites grew.

### 3.3. Pyrolysis Performance and High Temperature Tolerance

Thermogravimetric measurements were conducted on the pristine CNT film and the corresponding CNTF/MPC nanocomposites under the atmosphere of air and nitrogen, respectively. As shown in Figure 6, the pristine CNT film began to lose weight slightly at 217 °C in both air and nitrogen atmosphere, which was ascribed to the decomposition of organic molecules remaining on the surface of CNTs generated by the FCCVD preparation process. As shown in Figure 6c, for CNT film, the weight content of the organic component was 6.45% in an air atmosphere, which agreed well with the ratio of the peak area at about 275 °C in Figure 6d under a nitrogen atmosphere. Both indicate that the content of the small organic by-products for this FCCVD CNT film was about 6.45–6.53 wt.%. The TEM images of CNT film (Figure 7a) shows that the FCCVD CNT film was mainly composed of carbon nanotubes and a small amount of iron catalyst particles. In consideration of the oxidation reaction of the iron particles, the weight content of the carbon nanotube was estimated as 86.62 wt.%. Although the CNTF/MPC@1000 and CNTF/MPC@1300 both displayed the fastest degradation at 773 °C, typically, the peaks of amorphous carbon and crystalline carbon were at around 550 °C and 750 °C, respectively [45]. Therefore, the (derivative thermogravimetry) DTG curves of the two CNTF/MPC nanocomposites could be treated by deconvolution, where in the ratio of each peak area was utilized to estimate the weight content of the amorphous carbon, the nanotube, and the crystalline carbon, as shown in Table 2.

Figure 6a reveals that the 5% mass loss (Td_5_) of the CNT film took place at 358 °C, whereas that of CNTF/MPC@1000 and CNTF/MPC@1300 increased to 483 °C and 548 °C, respectively. Moreover, there was no DTG peak of organic components for CNTF/MPCs in Figure 6c. All these results indicate that during the preparation of the CNTF/MPC nanocomposites, the small organic molecules at the nanotubes surface decomposed, which happened simultaneously with the carbonization of the pitch matrix. Hence, the temperature resistance of the CNTF/MPC nanocomposites was significantly improved.

In light of the DTG peaks at 768 °C of CNTF/MPC@1000, the weight content of crystalline carbon was 15.13 wt.%. The content of crystalline carbon was considerably increased up to 29.24 wt.% in CNTF/MPC@1300, indicating that more amorphous carbon was transformed into crystalline carbon. The structure evolution further induced significant property improvements, e.g., the much higher tensile modulus and even higher volume density, as discussed above. Figure 6d demonstrates that the char yield of pristine CNT film was 91.69 wt.% due to the all-carbon structure of CNT, which ensured the structural stability of CNTF/MPC nanocomposites during the carbonization at high temperatures. Moreover, the mass retention rates of CNTF/MPC@1000 and CNTF/MPC@1300 at 1000 °C were 98.79 wt.% and 99.32 wt.%, respectively, in nitrogen. This demonstrates that the CNTF/MPC nanocomposites can maintain a high degree of integrity at temperatures up to 1000 °C without oxygen in potential applications.

Figure 7a reveals the TEM images of the pristine CNTs and CNT agglomerations. Figure 7c displays the cross-section morphologies of the CNT film. The CNT bundles showed an island pattern in the cross-section of the CNT network with ample space around them. The latter pores may serve as oxygen transport channels to accelerate the pyrolysis process of CNTs during the high temperature treatment. Figure 7b shows that the CNTs were completely coated by the carbon matrix. For the cross-section of CNTF/MPC nanocomposite, the FIB-SEM image in Figure 7d shows that the carbon matrix fully filled the pores within the CNT network. It could restrain the penetration of oxygen into the CNTF/MPC composites, which imparted better temperature tolerance. Figure 7b,d also verify that hot-pressing carbonization is a valuable technique to fabricate high-quality CNTF/MPC composites.

To further characterize the evolution of the amorphous carbon and the crystalline carbon in the CNTF/MPC nanocomposite, the XRD spectra of CNTF/MPC@1000 and CNTF/MPC@1300 were analyzed, as shown in Figure 8. The intensity of the amorphous peaks of CNTF/MPC@1000 was higher than that of CNTF/MPC@1300. Moreover, the (002) peak of CNTF/MPC@1300 had a stronger intensity than that of CNTF/MPC@1000, for which the full width at half maxima (FWHM) was accordingly 1.41 versus 2.39. These results strongly demonstrate that the massive amorphous carbon of the pitch base transformed into crystalline carbon during the carbonization treatment of CNTF/MPC@1300. Therefore, the value of Td_5_ of CNTF/MPC@1300 was much higher than that of CNTF/MPC@1000.

Figure 9 displays the DSC curves of MP and pre-impregnated CNTF/MP, which both revealed two exothermic peaks. According to the characterization results for MPC@1000 and MPC@1300 in Figure 4 and Figure 5, one may find that the first exothermic peak herein should have been associated with the generation of amorphous carbon and the structure transition from amorphous carbon to crystalline carbon. Likewise, the second exothermic peak further transformed from amorphous carbon into crystalline carbon. Hence, the curves could be divided by deconvolution into three peaks, as shown in Figure 9. The data are listed in Table 3. Comparing the peak positions of the MP, those of CNTF/MP all moved towards much lower temperatures. In addition, the values of the enthalpy change of CNTF/MP were all considerably higher than those of MP. For instance, the enthalpy changes of crystalline carbon at low temperature (LT) and at high temperature (HT) of CNTF/MP were 14.9 kJ/g and 2.7 kJ/g, respectively, which was more than two times and four times, respectively, than of MP. These results demonstrate that the CNTs serve a role of nucleation in the MP matrix, which promotes the carbonization process and the growth of micro-crystallites dramatically.

### 3.4. Conductivity Performance and the Temperature Dependence

Table 4 shows the electrical conductivities of the CNT film and the corresponding CNTF/MPC nanocomposites. The conductivity of the CNT film was 424 S/cm. Generally, it is determined by two factors: (a) the resistance of individual CNTs and (b) the contact resistance between CNTs. The conductivity of individual MWCNT reached 3.3 × 10^4^ S/cm [46], which was two orders of magnitude higher than that of its film assembly. Therefore, the contact resistance of CNTs played a key role in the conductive performance of the continuous CNT film. After the combination of the carbonized pitch matrix, the conductivity of CNTF/MPC@1000 increased to 683 S/cm, and CNTF/MPC@1300 displayed an even more improved conductivity of up to 841 S/cm. The results indicate that the contact resistance of inter-CNTs is significantly reduced by the fully infiltrated carbon matrix within the carbon tube networks. By comparing CNTF/MPC@1300 with CNTF/MPC@1000, it becomes evident that the higher carbonization temperature tended to generate a larger size and greater number of graphite crystallites. A larger proportion of carbon atoms form conjugated π bonds of SP^2^ hybrid, which increases delocalization of free electrons in carbon matrix. As a result, the electrical conduction between CNTs and MPC and internal CNT networks is enhanced. Therefore, the electrical conductivity of CNTF/MPC nanocomposites is superior to that of carbon family thin film materials. For instance, herein CNTF/MPC@1000 showed higher conductivity than pyrolytic carbon film of 500 S/cm [47], and even CNTF/MPC@1300 had higher conductivity than graphene paper (720 S/cm) [48]. As listed in Table 4, the thermal conductivity in the thickness direction of the CNT film was only 0.18 W/m∙K, whereas that of CNTF/MPC@1300 increased by an order of magnitude of 1.89 W/m∙K due to the heat transfer effect of the carbon matrix.

Figure 10 is a comparison plot of the electrical conductivities and volume densities of the CNTF/MPC nanocomposites with reported data of diverse CNT/C bulk composites [16,17,19,20,21,23,24,25,27,28,49]. The CNTF/MPC nanocomposites in this work showed relatively higher electrical conductivities than most CNT/C bulk composites presented in the literature. Liu et al. [17] added 5 wt.% CNTs to the high conductivity mesophase pitch powders (854 S/cm) to obtain a CNT/C composite with a conductivity of 1175 S/cm. However, the conductivity of the composite with 20 wt.% CNTs decreased sharply to 554 S/cm due to the serious aggregation of nanotubes. For the FCCVD CNT film, the entangled CNTs are always connected through catalysts building physical junctions among the network, which serve as continuous electrons transmission paths to enhance the electrical properties of the CNTF/MPC nanocomposites. By utilizing FCCVD CNT film, Zhang et al. [25] and Han et al. [50] both fabricated CNT/C composites with excellent electrical conductivities. However, the production process needs to be repeated at least three times to acquire the high-performance composites at a high time and energy cost. To take advantage of the good processability of FCCVD CNT film, the hot-pressing carbonization method without cycle steps was explored in this work to efficiently fabricate the CNTF/MPC nanocomposites. The optimized external pressure drove the pitch matrix fully impregnated into nano- and micro-spaces within the CNT film, rather than forming a shell only outside the film. The cladding conductive carbon matrix on the single nanotubes and their networks then effectively improved the electrical conductivities of the CNTF/MPC nanocomposites after carbonization.

The above studies reveal that CNTF/MPC nanocomposites have excellent high temperature resistance and electrical conductivity. Hence, the temperature dependence of the electrical resistance ranging from room temperature to 150 °C was investigated, as shown in Figure 11a, for the CNT film and the CNTF/MPCs. It shows that the temperature coefficient of resistance (TCR) of the CNT film was positive at 2.4 × 10^−3^ °C^−1^, whereas the TCRs of CNTF/MPC@1000 and CNTF/MPC@1300 were both negative at −1.8 × 10^−4^ °C^−1^ and −1.4 × 10^−4^ °C^−1^, respectively. As the temperature rose gradually, the entrapped air within the CNT networks expanded, leading to larger distances and higher contact resistances between CNTs. The free transferal of electrons became more difficult between nanotubes, which brought about a positive temperature coefficient effect in the range of 20–150 °C. After carbonization of the pitch matrix, the lock-up effect of carbon matrix eliminated the air thermal expansion in the CNTF/MPC nanocomposites. Hence, the TCR of the CNTF/MPCs was mainly determined by the contact resistance between the nanotubes and the carbon matrix. The TCRs of CNTF/MPC@1000 and CNTF/MPC@1300 both exhibited a negative tendency with the increasing temperature, which could be ascribed to the more active thermal motivations of the electrons.

To further reveal the temperature dependence mechanism of the CNTF/MPC nanocomposites, two types of electron conduction behavior were investigated: the variable range hopping conduction mechanism and the tunneling effect conduction mechanism [50], which can be expressed by the following two formulas, respectively:(1)$$\sigma T^{1/2}=\exp\left(-A/T^{1/4}\right)$$
(2)$$\sigma =\sigma_{0}\exp\left(-B/T^{1/2}\right)$$
where, σ is the electrical conductivity, *T* is the absolute temperature, σ_0_, *A* and *B* are constants.

The logarithm was taken for both sides of Equations (1) and (2) to draw the graphs, as shown in Figure 11b,c. The linear fitting coefficients of the hopping mechanism and the tunneling conduction mechanism both were very close to 1, suggesting these were mostly the electrical conduction behavior of the CNTF/MPCs. The defective structure of CNTs was the main reason to induce variable range hopping of free electrons. A portion of free electrons possibly hopped from one localized position of CNT to another or from one nanotube to another through the CNT defects. As the temperature rose, the jumps of the free electrons became more intense. Another quantity of free electrons, stimulated by thermal vibrations, was emitted from the one-dimensional channels of CNTs and transited through the barrier formed at the interface of the carbon matrix layer to the adjacent CNTs, leading to tunnel currents. Therefore, CNTF/MPC nanocomposites showed a negative temperature coefficient effect. Similarly, Vavro et al. [51] inserted C_60_ into CNTs, which also provided a negative TCR value of −2.8 × 10^−3^ °C^−1^. In contrast, the MP-based carbon matrix imparted higher temperature stability in electrical conductivity to the CNTF/MPC nanocomposites than the reported C_60_ effect.

## 4. Conclusions

This paper successfully fabricated a CNTF/MPC nanocomposite by incorporating continuous CNT film with mesophase pitch matrix. The FCCVD CNT film with great mechanical property and multi-functionality was selected as reinforcement. Though the efficient hot-pressing carbonization method, the carbon matrix infiltrated fully into the CNT networks to obtain high-quality and super-performance nanocomposites. After carbonization at 1300 °C, the density of CNTF/MPC increased to 1.53 g/cm^3^, showing tensile a strength and modulus of 119.8 MPa and 23.8 GPa, respectively. The mechanical properties were much higher than those of the pristine CNT film (45.5 MPa and 0.19 GPa, respectively). Interestingly, the CNTF/MPC nanocomposites displayed metallic luster after carbonization and they resisted cyclic bending without observable damage. The structure characterizations indicated that stacked aromatic layers in raw MP tended to rearrange into amorphous carbon and then transform to crystalline carbon during the carbonization process. A higher carbonization temperature always leads to a larger size and quantity of micro-crystallites in the MPC matrix. The thermal analysis proves that the CNTs serve as a role of nucleation in the MP, which accelerates the carbonization process and the formation of micro-crystallites significantly. Therefore, the prepared CNTF/MPC nanocomposites revealed excellent properties, e.g., the CNTF/MPC@1300 exhibited remarkably stable electrical conductivity of 841 S/cm at RT-150 °C with its thermal conductivity through a thickness direction of 1.89 W/m∙K, an order of magnitude higher than that of the CNT film. It had a mass retention of 99.32% at 1000 °C in nitrogen and a Td_5_ of 548 °C in air. In summary, the CNTF/MPC nanocomposites have superb multi-functional properties for potential applications in temperature resistance aerospace structures.

## Figures and Tables

**Figure 1 nanomaterials-11-00758-f001:**
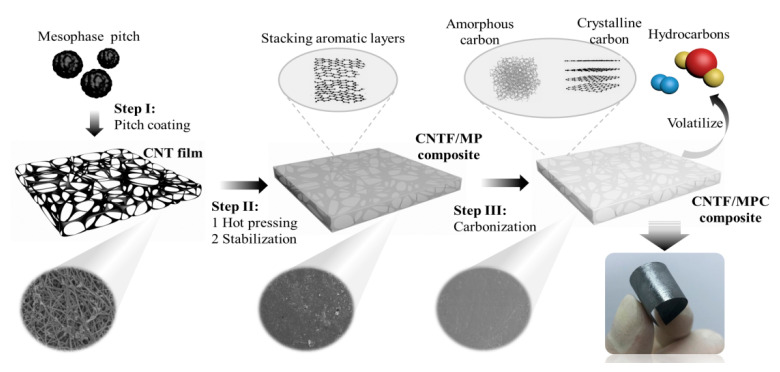
The preparation process of CNT film-reinforced pitch-based carbon nanocomposite by hot-pressing carbonization.

**Figure 2 nanomaterials-11-00758-f002:**
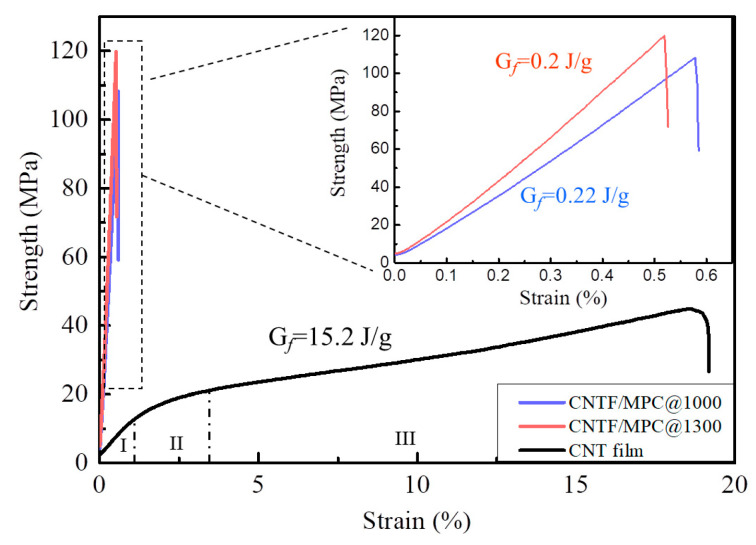
Typical stress–strain curves of CNT film and CNTF/MPC nanocomposites.

**Figure 3 nanomaterials-11-00758-f003:**
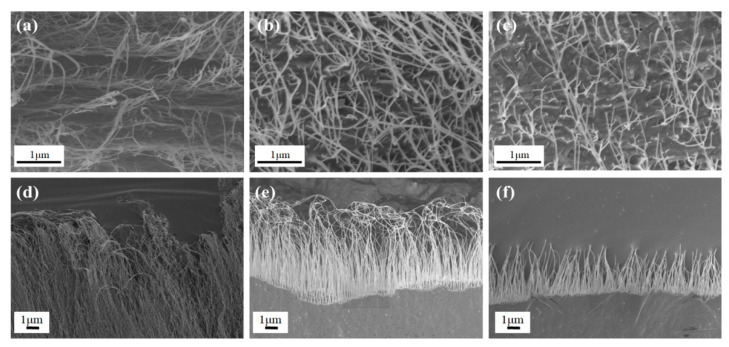
SEM images of the fracture morphologies of CNT film (**a**,**d**), CNTF/MPC@1000 (**b**,**e**), and CNTF/MPC@1300 (**c**,**f**).

**Figure 4 nanomaterials-11-00758-f004:**
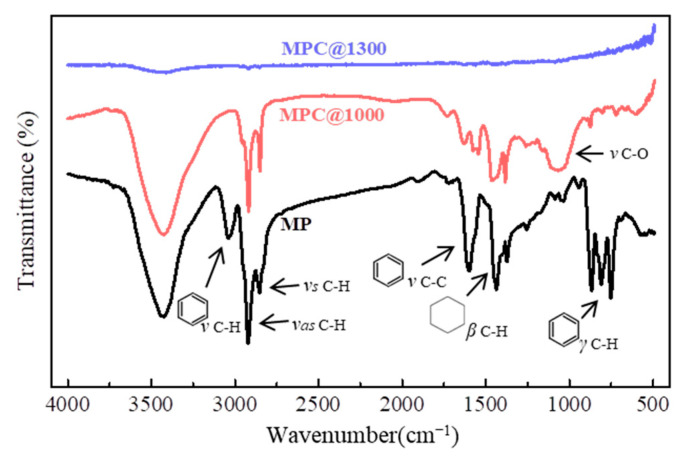
Infrared spectra of MP, MPC@1000, and MPC@1300.

**Figure 5 nanomaterials-11-00758-f005:**
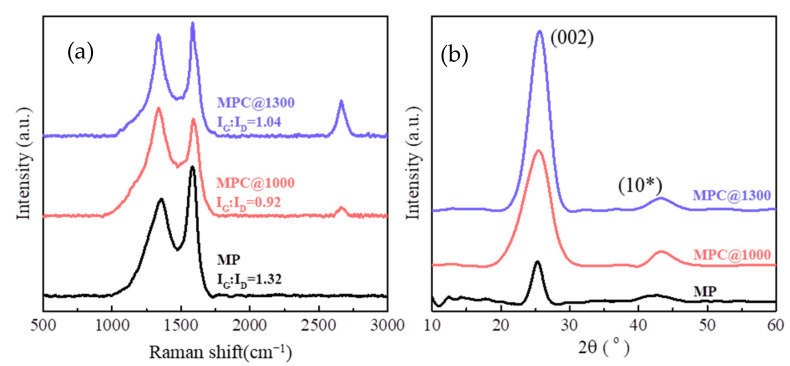
(**a**) Raman spectra and (**b**) XRD spectra of MP, MPC@1000, and MPC@1300.

**Figure 6 nanomaterials-11-00758-f006:**
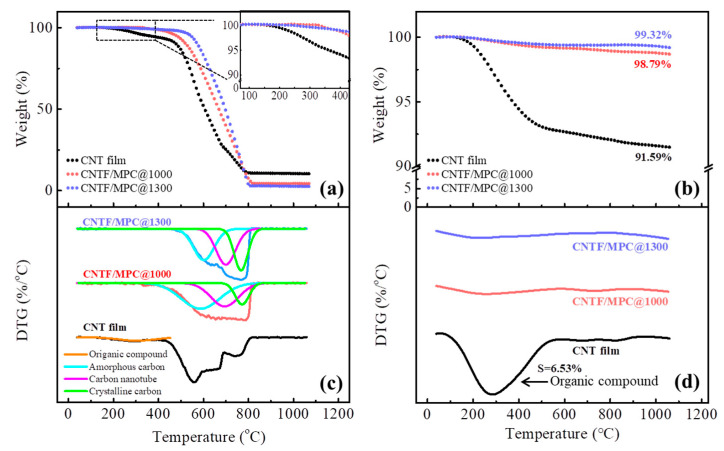
TGA and DTG curves of CNT film, CNTF/MPC@1000, and CNTF/MPC@1300 in air (**a**,**c**) and in nitrogen (**b**,**d**).

**Figure 7 nanomaterials-11-00758-f007:**
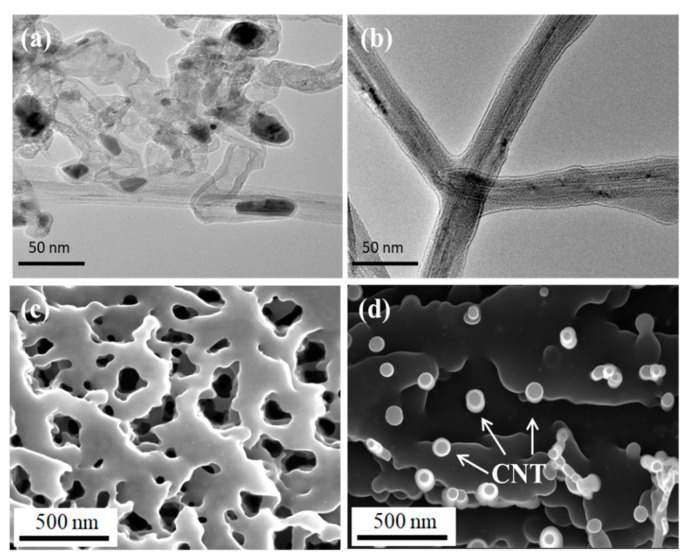
TEM images and the cross-sectional FIB-SEM images of the CNT film (**a**,**c**) and CNTF/MPC nanocomposites (**b**,**d**).

**Figure 8 nanomaterials-11-00758-f008:**
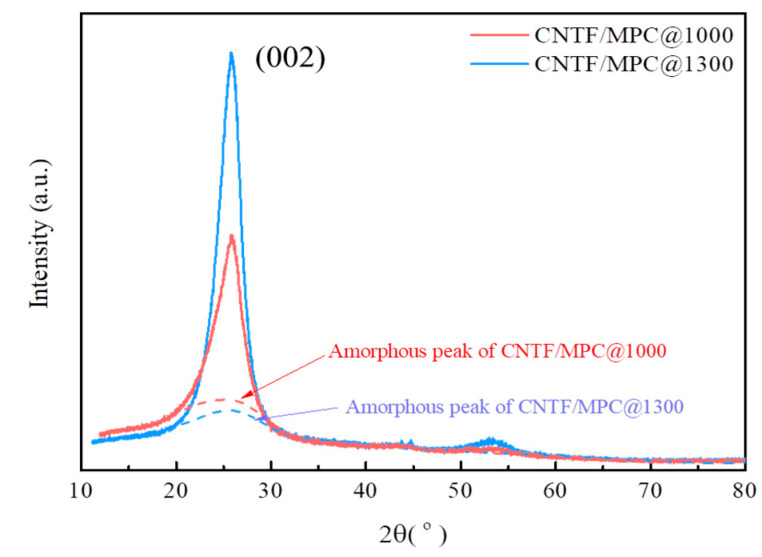
XRD patterns of CNTF/MPC@1000 and CNTF/MPC@1000.

**Figure 9 nanomaterials-11-00758-f009:**
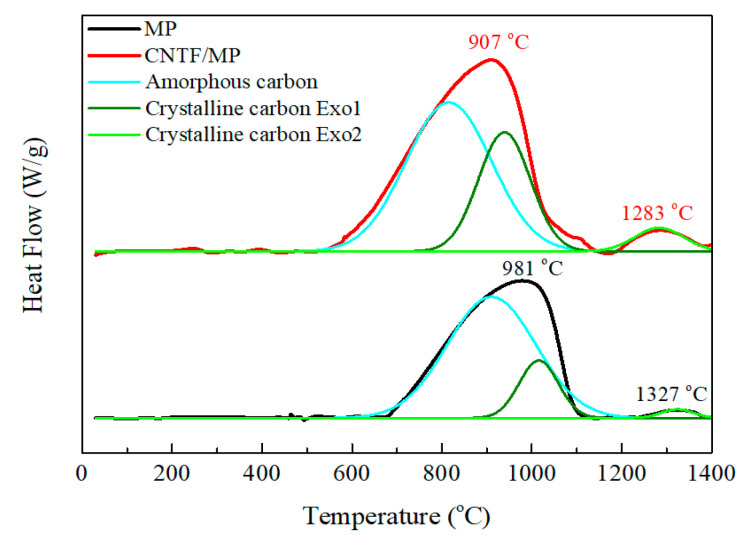
DSC curves of MP and CNTF/MP composites.

**Figure 10 nanomaterials-11-00758-f010:**
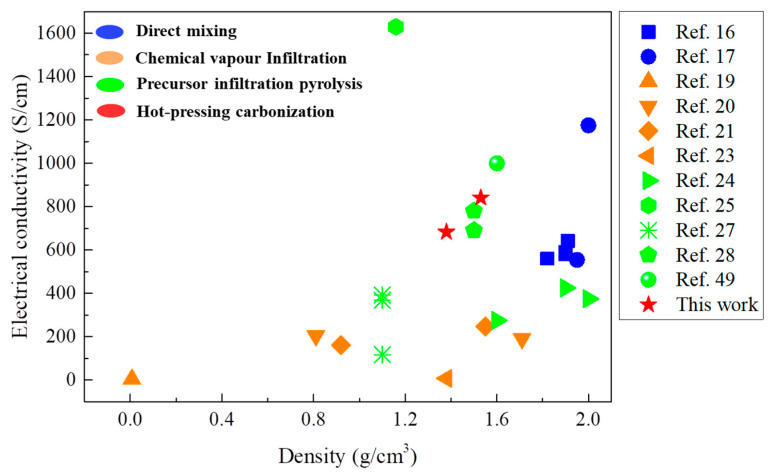
Plots of electrical conductivity of the studied CNTF/MPC nanocomposites, in comparison with the reported values of different CNT/C bulk composites.

**Figure 11 nanomaterials-11-00758-f011:**
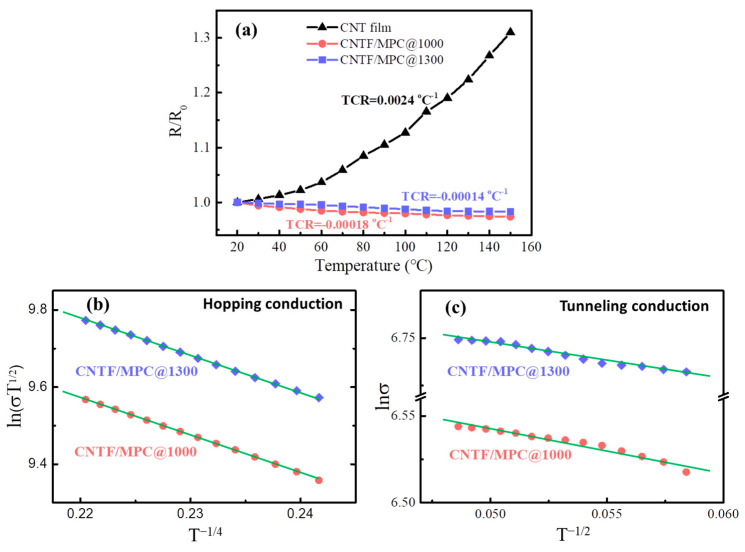
(**a**) Temperature dependence of resistance change in CNT film and CNTF/MPC nanocomposites, (**b**) fitting line of variable range hopping conduction mechanism, and (**c**) tunneling conduction mechanism.

**Table 1 nanomaterials-11-00758-t001:** The calculated structural parameters of different specimens from the XRD spectra.

Sample	2θ (degree)	FWHM (degree)	d(002) ^a^ (nm)	L_C_(002) ^b^ (nm)
MP	25.3	1.79	0.352	5.05
MPC@1000	25.5	4.25	0.349	2.13
MPC@1300	25.6	3.14	0.348	2.88

^a^ d_002_ = λ/(2sinθ_002_), ^b^ L_C_ = λ/(βcosθ_002_). Herein, λ, β, and θ are the wavelength of the X-ray, FWHM (full width at half maxima), and the peak position of (002), respectively.

**Table 2 nanomaterials-11-00758-t002:** The mass percentage of different components of CNT film and CNTF/MPC nanocomposites.

Specimen	Mass (%)			
	Organic Component	Amorphous Carbon	Carbon Nanotube	Crystalline Carbon
CNT film	6.45	-	86.62	-
CNTF/MPC@1000	-	51.21	33.64	15.13
CNTF/MPC@1300	-	33.08	35.53	29.24

**Table 3 nanomaterials-11-00758-t003:** The data of DSC curves of MP and CNTF/MP as shown in Figure 9.

		MP		CNTF/MP	
		Peak Position (°C)	Enthalpy Change (kJ/g)	Peak Position (°C)	Enthalpy Change (kJ/g)
Amorphous carbon		904	25.8	815	30.5
Crystalline carbon	LT ^a^	1014	5.4	938	14.9
	HT ^b^	1327	0.6	1283	2.7

The enthalpy changes of crystalline carbon ^a^ at low temperature and ^b^ at high temperature.

**Table 4 nanomaterials-11-00758-t004:** The electrical and thermal conductivities of CNT film and CNTF/MPC nanocomposites.

Sample	σ ^a^ (S/cm)	Increment (%)	α ^b^ (mm^2^/s)	λ ^c^ (W/m∙K)	Increment (%)
CNT film	424	-	0.54	0.18	-
CNTF/MPC@1000	683	61	1.5	1.49	727
CNTF/MPC@1300	841	98	2.26	1.89	950

^a^ σ is electrical conductivity; ^b^ α is thermal diffusivity; ^c^ λ is thermal conductivity.

## Data Availability

The data presented in this study are available on request from the corresponding author.

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
