# Peer review of "The Fabrication and Properties of a Bendable High-Temperature Resistance Conductive Pitch-Based Carbon/CNT Film Nanocomposite"

_nanomaterials, 2021, doi:10.3390/nano11030758_

Round 1

Reviewer 1 Report

Ref.comments to the paper nanomaterials-1140555 titled as “The Fabrication and Properties of a Bendable High-tempera-2ture Resistance Conductive Pitch-based Carbon/CNT Film 3Nanocomposite”, written by Zhe Che,ShaokaiWang, Yizhuo Gu, Wei Zhang, Cai Jiang and Min Li.

It is well known, that because at the present time the optoelectronics area is extended and occupied so many directions, the study of the novel composites with the unique features is modern.

From this point of view the paper is interesting, useful and actual.

For the first, it is remarkable that the authors have made a decent literary search consisting of 46 references. This indicates the knowledge of the problem and finding ways to solve it.

The authors have shown the preparation process of the reinforced carbon film in details; presented the typical stress–strain curves of CNT film and modified CNTF/MPC nanocomposites; they have revealed SEM images of the fracture morphologies of CNT film and  CNTF/MPC structures; they have obtained the differences in the infrared, Raman spectra and DSC curves of MP and CNTF/MP composites.

Interesting results have been shown in Table 1. - calculated structural parameters of different specimens from XRD patterns; in Table 2. - mass percent of different components of CNT film and CNTF/MPC nanocomposites; in Table 3. - data of DSC curves of MP and CNTF/M; in Table 4. - electrical and thermal conductivities of CNT film and CNTF/MPC nanocomposites.

Useful results have been presented in Figure 8. - Plots of electrical conductivity of the studied CNTF/MPC nanocomposites, in comparison with the reported values of different CNT/C bulk composite. These data have been presented in comparison with that obtained by other scientific teams. Thus, the results shown can extend your knowledge in this direction.

The conclusion part collects the basic results shown in the paper.

Some little recommendation and questions:

  1. I would like to ask the author about the refractive parameters change in your novel composite. It is well known that the CNT refractive index is close to 1,05-1,1. What about your new composite refractive index?
  2. Parameter T in formula (1, left part) should be tilted as the Latin symbol.
  3. Please add in your ref.list some papers published on 2018-2020 years.

So, the article is good. In my local opinion, this paper can be published in the Journal with the little corrections.

Author Response

We are very thankful to the editor and the reviewers for their deep and thorough review. Now we have revised our manuscript in the light of their helpful suggestions and comments. The response and revisions were listed in the attachment. 

Reviewer 2 Report

The study by Che et al. covers interesting though explored in the literature compounding of CNTs with mesophase pitch - toward enhanced mechanical and electrical performance. The main issue of the paper is nevertheless related with a rather poorly crystalline CNT film serving as the to-be-infiltrated support. In fact, there is a lacking cross-characterization of this nanomaterial. Also, the rationale behind using shorter, largely defectuous CNTs is missing. The final outcome cannot be therefore fully competitive - as authors show themselves in Fig. 8. Could authors address this problem and re-elaborate the clear advantages over the so-far existing solutions - particularly in Abstract, Results and Conclusions? This correction would be of significant importance for the sub-field.

Author Response

(The authors gave the same response as above.)

Round 2

Reviewer 2 Report

The concerns were applied correctly and accordingly. The paper can be accepted in its current form.